:ᐧ PLOS ONE

# Divulging diazotrophic bacterial community structure in Kuwait desert ecosystems and their N$_2$-fixation potential

M. K. Suleiman, A. M. Quoreshi*, N. R. Bhat, A. J. Manuvel, M. T. Sivadasan

Desert Agriculture and Ecosystems Program, Environment and Life Sciences Research Center, Kuwait Institute for Scientific Research, Safat, Kuwait

* aquoreshi@kisr.edu.kw

**Data Availability Statement:** All relevant data are within the manuscript and its Supporting Information files.

## Abstract

Kuwait is a semi-arid region with soils that are relatively nitrogen-poor. Thus, biological nitrogen fixation is an important natural process in which N$_2$-fixing bacteria (diazotrophs) convert atmospheric nitrogen into plant-usable forms such as ammonium and nitrate. Currently, there is limited information on free-living and root-associated nitrogen-fixing bacteria and their potential to fix nitrogen and aid natural plant communities in the Kuwait desert. In this study, free living N$_2$-fixing diazotrophs were enriched and isolated from the rhizosphere soil associated with three native keystone plant species; *Rhanterium epapposum*, *Farsetia aegyptia*, and *Haloxylon salicornicum*. Root-associated bacteria were isolated from the root nodules of *Vachellia pachyceras*. The result showed that the strains were clustered in five groups represented by class: γ-proteobacteria, and α-proteobacteria; phyla: Actinobacteria being the most dominant, followed by phyla: Firmicutes, and class: β-proteobacteria. This study initially identified 50 nitrogen-fixers by16S rRNA gene sequencing, of which 78% were confirmed to be nitrogen-fixers using the acetylene reduction assay. Among the nitrogen fixers identified, the genus *Rhizobium* was predominant in the rhizosphere soil of *R. epapposum* and *H. salicornicum*, whereas *Pseudomonas* was predominant in the rhizosphere soil of *F. aegyptia*, The species *Agrobacterium tumefaciens* was mainly found to be dominant among the root nodules of *V. pachyceras* and followed by *Cellulomonas*, *Bacillus*, and *Pseudomonas* genera as root-associated bacteria. The variety of diazotrophs revealed in this study, signifying the enormous importance of free-living and root-associated bacteria in extreme conditions and suggesting potential ecological importance of diazotrophs in arid ecosystem. To our knowledge, this study is the first to use culture-based isolation, molecular identification, and evaluation of N$_2$-fixing ability to detail diazotroph diversity in Kuwaiti desert soils.

## Introduction

Symbiotic bacterial association with plants mediates most of the nitrogen fixation process in terrestrial ecosystems. However, non-symbiotic N$_2$ fixation by free-living nitrogen-fixing

**Funding:** This work was supported by the Kuwait Foundation for the Advancement of Sciences; Kuwait Institute for Scientific Research, P215-42SL-01; 019-190033 (http://www.kfas.org/; http://www.kisr.edu.kw/en/) to MKS. The funder had no role in study design, data collection and analysis, decision to publish, or preparation of the manuscript.

**Competing interests:** The authors have declared that no competing interests exist.

bacteria harboring in soil systems can considerably contribute to the nitrogen fixation pool in various ecosystems, particularly in desert ecosystems [1, 2]. Nitrogen is an essential element required for plant growth and development is considered a limiting factor in plant productivity [3, 4], and can affect the life of microbes and other living organisms [5]. Nitrogen is an important component of amino acids, the building blocks of proteins; chlorophyll, the green pigment required for photosynthesis; ATP, the primary energy carrier; nucleic acids, the genetic material of living organisms [6]. Thus, nitrogen plays a fundamental role in the growth and productivity of crops. Nitrogen is abundant in the atmosphere (almost 78%) as molecular nitrogen but cannot be utilized directly by the plants. Therefore, biological N$_2$ fixation is necessary to convert elemental nitrogen into ammonia, which is readily available to bacteria and plants. Nitrogen fixation is an essential step in the global nitrogen cycle as it restores and recompenses the overall nitrogen lost because of denitrification [7]. Atmospheric nitrogen is fixed into the soil, utilized by the plants, and returned to the soil and atmosphere through a process called the nitrogen cycle. In general, nitrogen cycle is a four-step process and comprises, nitrogen fixation (conversion of atmospheric di-nitrogen (N$_2$) to nitrate or ammonia), nitrification (conversion of ammonia to nitrates), assimilation (incorporation of the nitrates in to the plant tissues), and denitrification (conversion of nitrate present in the dead plants to dinitrogen).

Diazotrophs are a specialized group of bacteria that capable of biological nitrogen fixation by utilizing their nitrogenase system which converts atmospheric di-nitrogenin to readily available fixed nitrogen [1]. Diazotrophs can live freely in the soil, like *Pseudomonas*, *Azotobacter*, and *Cyanobacteria*, or can establish symbiotic association with certain species, like *Rhizobium* and *Bradyrhizobium*). These diazotrophs or biological nitrogen fixers associated with the plants and rhizosphere are possible alternatives to inorganic nitrogen fertilizer and supports the growth and productivity of the plants sustainably [8]. Inoculation of biological nitrogen fixers increased nitrogen fixation in *Phaseolus vulgaris* L. [9]. In addition, similar growth was reported in *Oryza sative* L (rice crop) when inoculated with plant growth promoting rhizobacteria plus half fertilization and full fertilization without inoculation [10]. It has been suggested that the use of phosphate-solubilizing diazotrophs is a good strategy to promote phosphate solubilization and / or nitrogrn use efficiency in rice plants [11]. Although diazotrophs play a key role in making nitrogen sources available to plants, little is known about the communities associated with soil, rhizosphere, and endosphere, and their importance in arid ecosystems [1, 12].

Arid regions are one of the harshest regions on this planet and present huge challenges in maintaining vegetation growth and plant productivity. The nutrients required for plant growth are lacking in the Kuwaiti desert soil due to scarce organic matter (<1%), low clay materials, high calcareous materials, low essential nutrients [13], poor precipitation and extreme moisture deficiency [14, 15]. Availability of soil nutrients is important for successfully restoring the degraded soil. Microorganisms and their bioactivity could play an essential role in nutrient cycling and improvement of soil fertility in arid terrestrial ecosystems. Despite the prevalence of severe environmental conditions in the desert low rain fall, extremely high temperatures, high level of solar radiation, scarcity of nutrients, and high salinity a broad range of organisms including plants, have adapted to the extreme conditions by developing different adaptive mechanisms [16]. It is believed that the desert microbiome may be a key factor for desert plants to adapt to these extreme conditions [17–19]. Microbes present in desert ecosystems are believed to be proficient in enhancing plant growth and stress tolerance and play an essential roles in nutrient cycling [20, 21]. Therefore, prudent management of soil resources for sustainable productivity is required for improving the nutritional condition of arid soils. The use of beneficial bio-inoculants could be one such approach. Therefore, there is a need to identify

N$_2$-fixing bacteria that can be used as bio-inoculants as an alternative to inorganic chemical fertilizers, for plants to grow and develop sustainably in the desert. The approach is considered an environmentally safe strategy to improve the quality of the soil without introducing chemical fertilizers. To our knowledge, no data exist on indigenous nitrogen-fixing bacterial communities from Kuwaiti desert soils. This study reports for the first time on the isolation and characterization of free-living and rhizobacterial communities associated with rhizosphere soils of few native shrubs and roots of *Vachellia pachyceras* from the Kuwaiti desert.

The aim of this study was to isolate, screen, and identify free-living and root-associated diazotrophic communities present in the rhizosphere soil of economically important Kuwaiti native plants, *Rhanterium epapposum*, *Farsetia aegyptia*, and *Haloxylon salicornicum* and root nodules of *V. pachycras*. Furthermore, an effort was undertaken to evaluate nitrogen-fixing ability of the isolated bacterial strains by the acetylene reduction assay (ARA).

## Materials and methods

### Sampling

Soil from the rhizosphere zone (soil around the plant root zone) of *R. epapposum*, *F. aegyptia*, and *H. salicornicum* located at KISR's Station for Research and Innovation (KSRI; GPS: N 29˚ 09.904'; E 047˚ 41.211'), Sulaibiya, were selected for the isolation of free-living nitrogen fixing bacteria. The selected plant were extracted carefully using a shovel and the root ball of the uprooted plant was shaken gently inside a sterile zip lock bag, to collect the rhizosphere soil. Rhizosphere soil samples were collected in triplicate from each plant species, as previously described. Similarly, root samples with nodules were collected from *V. pachyceras* from different locations inside KSRI, Sulaibiya. The lateral roots of *V. pachyceras* with root nodules were identified by digging around the plant and cut carefully with pruning scissors. Excess soil was removed and the root with the root nodules was transferred to a sterile zip lock bag. All materials used for sample collection were surface sterilized with 70% ethanol prior to use. The collected soil and root samples were transported to the laboratory on ice in a cooler box.

### Isolation and primary screening of diazotrophic bacteria

An enrichment culture method, using nitrogen-free semi-solid malate media and nitrogen deficient malate media with bromothymol blue (BTB) indicator [22] was adopted for the isolation of putative free-living nitrogen-fixing bacteria from 1 g of rhizosphere soil of *R. epapposum*, *F. aegyptia*, and *H. salicornicum*. A modified procedure was used for the isolation of *Rhizobium* on yeast mannitol agar medium containing 0.0025% Congo red from 100 mg root nodules of *V. pachyceras* [23]. Root nodules were surface sterilized using 95% (v/v) ethanol for 10 s and washed at least 7 times with sterilized distilled water. The plates were incubated at 28˚C for 3 to 7 days, and single colonies of putative N$_2$-fixers were carefully selected and re-streaked on yeast mannitol agar media for further purification. The procedure was repeated and the pure culture of all putative N$_2$-fixers were stocked in yeast mannitol broth with 15% glycerol and stored in -80˚ C freezer. Slants were prepared for the pure cultures and kept in the refrigerator for day-to-day experiments. All bacterial isolates were labeled with the first three letters of the plant species-source of isolation-serial number (e.g. Rha-S-1 [*R. epapposum*-soil-bacteria no.1]).

Primary screening of pure cultures to identify potential N$_2$-fixers was done by measuring the blue zone formed by the isolates when grown on nitrogen-deficient malate media with BTB. A color change to blue was considered as a positive indication for the presence of nitrogen-fixers [24].

## Biochemical identification of the isolates

Biolog® GEN III Microbial Identification System (Biolog Inc., Hayward, CA, USA) was used for the biochemical identification of putative N$_2$-fixers from the isolates of rhizosphere soil and root nodules of test species. The system identifies microorganisms based on their ability to metabolize all major classes of biochemical compounds and determines important physiological properties, such as pH, salt and lactic acid tolerance, reducing power, and chemical sensitivity. Pure bacterial isolates were streaked on biolog universal growth (BUG) agar medium and incubated for 24 hr at 30°C. Subsequently, individual colonies were suspended in IFA inoculating fluid such that the cell density was in the range of 90–98% as per the recommendation of Biolog Inc. A total of 100 μL of the prepared suspension was transferred to each well of a 96-well microplate, which was incubated in an OmniLog incubator at 33°C for 24 h. Colorimetric response was measured using Biolog OmniLog system and compared with the Biolog® database for identity confirmation (GEN III database and characteristics v2.7).

## Acetylene reduction assay (ARA)

Bacterial strains were grown in modified Fraquil medium, a nitrogen free medium based on Fraquil medium which contains $5 \times 10^{-3}$ M KH$_2$PO$_4$, $2.3 \times 10^{-3}$ M K$_2$HPO$_4$, $6.8 \times 10^{-4}$ M CaCl$_2$, $4.05 \times 10^{-4}$ M MgSO$_4$ (7H$_2$O), $10^{-8}$ M CuCl$_2$ (2H$_2$O), $2.25 \times 10^{-7}$ M MnCl$_2$ 4(H$_2$O), $2.43 \times 10^{-8}$ M CoCl$_2$ (6H$_2$O), $5.3 \times 10^{-8}$ M ZnSO$_4$ (7H$_2$O), $5 \times 10^{-7}$ M MoO$_4$Na$_2$, $5 \times 10^{-6}$ M FeCl$_3$ (7H$_2$O), $10^{-4}$ M EDTA, $5.5 \times 10^{-2}$ M Glucose, and $5.48 \times 10^{-2}$ M D-Mannitol, pH 6.7. Bacteria were inoculated into the modified Fraquil medium, containing with 3% agar using sterile and disposable spreaders and incubated at 28°C in a thermoregulated incubator (INFORS-HT Multitron, Switzerland).

The nitrogen-fixing ability of all the isolates found positive in the primary screening was determined by ARA. In this procedure, ethylene produced by the bacterial culture was quantified and the results were expressed in nano moles of C$_2$H$_4$ produced h$^{-1}$ culture$^{-}$1. Bacteria growing in petri dishes were collected after 1, 3 and 7 d of incubation. Dishes were transferred to 250 mL glass jars. The jars were closed with a lid equipped with a septum for gas sampling. Fifteen percent of the headspace was replaced by acetylene (C$_2$H$_2$) which was produced when calcium carbide reacted with water (according to the reaction: CaC$_2$ + 2H$_2$O Ca (OH)$_2$ + C$_2$H$_2$) in a Tedlar bag. Petri dishes were incubated in the presence of C$_2$H$_2$ for 6 h. A total of 3mL of headspace was collected after 3 h and 6 h of incubation and, transferred to 3 mL vacuum vials. Ethylene concentration was then qualified by gas chromatography using Shimatzu 8A (Shimatzu, Japan) equipped with a flame ionization detector (GC-FID). A control N$_2$ fixer *Azotobacter vinelandii*, was grown and assayed under similar conditions for comparison. In addition to modified Fraquil agar, the test bacterial strains were grown in yeast mannitol broth, agar and modified Fraquil broth and assayed for acetylene reduction under similar conditions.

After conducting the acetylene reduction assay (on day 1, 3 and 7), bacterial density was estimated for each petridish to normalize ARA data to cell density (OD). This allows for a better comparison of N$_2$ fixation potential between species. Cells were suspended by adding 2 mL of culture medium to the dishes and disturbing bacterial mats using an inoculation spreader. The solution was then transferred to a 15-mL tube. The procedure was repeated until no visible bacterial mats remained at the surface of the dish. Cells were then pelleted by centrifugation and the supernatant removed. Thereafter, the cells were resuspended in 1 mL of medium and OD (620 nm) was measured by UV-vis spectrometry. Cell suspension was diluted using the culture medium when needed (saturation of the UV detector).

## Molecular identification for the isolates

Aliquots from the fresh culture were picked with a sterile transfer pipet and resuspended in 200 μL of 45 mg/mL lysozyme solution (cat. No. L4919 –Millipore-Sigma Darmstadt, Germany). The cell suspension was incubated at 37 $^o$C for 30 min, as recommended by the manufacturer. Concentration of eluted DNA was quantified with a Nanodrop 2000 (Thermo Fisher, Waltham, MA, USA). Polymerase chain reactions (PCR) was performed on the eluted DNA using bacterial universal primers 358F and 907R specific for bacterisl 16S rRNA gene. PCR conditions were the same for both 358F (CTACGGGAGGCAGCAG [25]) and 907R (CCGTCAATTCMTTTRAGTTT [26]). The 25 μL reaction mix consisted of 1 μL bacterial DNA, 1 U of platinum Taq (Invitrogen), 2.0 mM MgCl$_2$, and 0.2 mM dNTP mix. The following thermocycle program was used for amplification: 95°C for 30 seconds; followed by 30 cycles of 95°C, 54°C and 72°C for 50 seconds each; and extension at 72°C for 5 min, using MJ Research PTC-225 Peltier Thermal Cycler. Amplifications were visualized on a 1.0% agarose gel electrophoresis (TAE) and amplicons were sequenced using the Sanger sequencing method with two 16-capillary genetic analyzers 3130XL (Applied Biosystems).

DNA sequences were edited with BioEdit version 7.0.5 [27, 28]. The BLASTn algorithm [29] was used to query for highly similar sequences using the bacterial 16S ribosomal RNA RefSeq Targeted Loci Project. Sequences were aligned using ClustalX version 1.81 [30]. Phylogenetic analyses were performed in order to identify and/ or place unknown sequence in the taxonomic tree of known species. Phylogenetic analyses were conducted using MEGA 7.0.21 software [31, 32]. First, phylogenetic analyses were realized using the neighbor-joining (NJ) method [33] based on the Kimura 2-parameter method [34] to visualize the approximate tree topology. Subsequently, the maximum likelihood (ML) method, based on the Kimura 2-parameter model [34], was used to construct the final tree. Initial tree(s) for the heuristic search were obtained automatically by applying NJ and BioNJ algorithms to a matrix of pairwise distances estimated using the maximum composite likelihood (MCL) approach, and then selecting the topology with superior log likelihood value. All analyses were bootstrapped 1000 times [35]. Bayesian inference of phylogeny was calculated using MrBayes program, assuming a 4 X 4 model and non-variable substitution rates among sites–gamma rates. Analyses were based on two runs of four Markov chain Monte Carlo analyses where 2, 000, 000 generations were generated, burning fraction at 0.5 rate and sampled every 100 generations for a total of 10, 000 trees generated [36].

## Results

### Isolation and primary screening of diazotrophic communities

Free-living nitrogen-fixing diazotrophs were isolated from rhizosphere soils of three native shrubs and root nodules were isolated from one tree species. A total of 10, 13, and 22 morphologically different pure bacterial strains were isolated from the rhizosphere soil of *R. epapposum*, *F. aegyptia*, and *H. salicornicum*, respectively. In contrast, 61 pure cultures with different morphotypes were isolated from the root nodules of *V. pachyceras*. The results indicates that several other species of free-living nitrogen-fixing bacteria may be present in the desert soil of Kuwait around the selected plant species.

For the primary screening of diazotrophic bacteria, nitrogen-free malate media with BTB as an indicator was used. The diameter of the blue colored zone for each bacterial isolate was recorded. Approximately, 70, 77, and 73% of the bacterial isolates were obtained from the rhizosphere soil of *R. epapposum*, *F. aegyptia*, and *H. salicornicum*, respectively, which were found to have a potential for N$_2$ fixation. However, around 38% of the bacterial isolates from

the root nodule of *V. pachyceras* were found to have a potential for nitrogen fixation. Diameter of blue zones ranged from 0.1 cm to 3.6 cm (Table 1).

## Acetylene reduction assay (ARA)

Approximately, 78% of the primary screened bacterial isolates grew efficiently in modified Fraquil agar medium (Figs 1–4, Table 1). The diazotrophic nature of the primary screened isolates was determined by ARA. About 39 isolates exhibited N$_2$-fixing ability, but the level of ethylene production (C$_2$H$_4$) ability of the isolates varied considerably with different isolates (Figs 1–4) and was characterized as low maximum, moderate maximum, high maximum, and very high maximum N$_2$-fixing ability (Table 1). However, no direct relationship was established between the diameter of the blue zone measured in the primary screening and the levels of N$_2$-fixation potential detected by ARA. The results suggest that ARA is the most reliable method for evaluating nitrogen fixing ability of N$_2$-fixers. ARA not only evaluated the nitrogenese activity of the bacteria but also quantified the nitrogen fixation ability. The amount of ethylene produced in ARA system is expressed in nano moles of ethylene (C$_2$H$_4$) produced per hour related to cell density measured per milliliter. In the ARA tests, the level of bacterial N$_2$-fixed correlated to conversion of acetylene (C$_2$H$_2$) to ethylene (C$_2$H$_4$), varied with different isolates and incubation time, and ranged among the strains associated with different rhizosphere soils and root samples from 7 to 145 nmol/h/OD (Figs 1–4). For comparison, the ethylene production rates of the well-known free-living N$_2$ fixer *Azotobacter vinelandii*, grown and assayed under similar conditions, was evaluated at 500 to 3,000 nmol/h/OD.

Interestingly, none of the cultures grew in the liquid modified Fraquil medium. This may be because the cultures were oxygen stressed, due to agitation (180 rpm) process. Oxygen is a known inhibitor of nitrogenase, which is responsible for N$_2$-fixation. Static cultures could yield better growth performances. However, static cultures could develop biofilms making it difficult to measure bacterial growth accurately. The attempt of screening for N$_2$-fixation ability of the isolated strains grown on yeast manitol agar and broth media shows that none of the bacterial strain achieved significant N$_2$-fixation activity. This is likely due to the presence of nitrogen contamination in the medium.

## Biochemical and molecular identification of the isolates

Of the 50 isolates screened after primary screening, 27 were successfully identified as bacterial strains from the rhizosphere soil of *R. epapposum*, *F. aegyptia*, *H. salicornicum* and root nodules of *V. pachyceras*, using the Biolog® GEN III Microbial Identification System. The isolates belonged to *Rhizobium*, *Pseudomonas*, *Bacillus*, *Enterobacter*, *Burkholderia*, *Macroccoccus*, *Microbacterium*, and *Advenella*. *Rhizobium* and *Pseudomonas* were the dominant genera with the species *Rhizobium radiobacter*, *Rhizobium rhizogenes* among the genera *Rhizobium* and *Pseudomonas stutzeri*, *Pseudomonas viridilivida*, and *Pseudomonas fluorescens* among the genus *Pseudomonas*. The second dominant genera were *Bacillus* and *Enterobacter* and the species identified were *Bacillus megaterium*, *Bacillus odysseyi*, and *Bacillus simplex/butanolivorans* for *Bacillus* and *Enterobacter cowanii* and, *Enterobacter homaechei* for *Enterobacter*. The least dominant species were *Burkholderia anthian/caribensis*, *Macroccoccus equipercicus*, *Microbactetium* spp. (CDC.A-4), and *Advenella incenata* (Fig 5A). Using the biochemical identification system, a single bacterial species called *E. cowanii* was found in the rhizosphere soil of *R. epapposum*, whereas *Pseudomonas* spp., *A. incenata* was found in the rhizosphere soil of *F. aegyptia* and *P. stutzeri*, *M. equipercicus* was found in the rhizosphere soil of *H. salicornicum*. The different bacterial species observed in the root nodules of *V. pachyceras* were *R. radiobacter*, *R. rhizogenes*, *P. viridilivida*, *P. fluorescens*, *B. megaterium*, *B. odysseyi*, *B. simplex/butanolivorans*, *B.*

**Table 1. Diameter of zone of colorization developed on nitrogen free malate media and nitrogen fixation potential of the bacterial isolates isolated from rhizospheric soil of selected native plants and root nodule of *Vachellia pachyceras*.**

| Isolate ID Number | Diameter of Zone of Colorization (cm) | Relative Nitrogen Fixation Potential (Acetylene Reduction Assay) | Isolate ID Number | Diameter of Zone of Colorization (cm) | Relative Nitrogen Fixation Potential (Acetylene Reduction Assay) |
|---|---|---|---|---|---|
| Rhizospheric Soil of *Rhanterium epapposum* | | | | | |
| RhaS-1 | 0.1 | ++++ | RhaS-6 | 0.6 | ++ |
| RhaS-2 | 0.3 | +++ | RhaS-8 | 0.0 | - |
| RhaS-3 | 1.1 | L | RhaS-9 | 0.8 | ++++ |
| RhaS-4 | 0.8 | ++ | RhaS-10 | 0.0 | - |
| RhaS-5 | 0.0 | - | RhaS-12 | 1.0 | +++ |
| Rhizospheric Soil of *Farsetia aegyptia* | | | | | |
| FarS-1 | 1.5 | +++ | FarS-8 | 2.7 | ++++ |
| FarS-2 | 2.1 | D | FarS-9 | 1.3 | ++++ |
| FarS-3 | 1.3 | D | FarS-10 | 0.1 | L |
| FarS-4 | 1.6 | D | FarS-11 | 0.0 | - |
| FarS-5 | 1.5 | L | FarS-12 | 0.0 | - |
| FarS-6 | 0.1 | D | FarS-13 | 0.0 | - |
| FarS-7 | 0.1 | D | - | - | - |
| Rhizospheric Soil of *Haloxylon salicornicum* | | | | | |
| HalS-1 | 2.2 | ++ | HalS-14 | 2.2 | +++ |
| HalS-2 | 0.0 | - | HalS-15 | 0.0 | - |
| HalS-3 | 2.3 | + | HalS-16 | 1.6 | L |
| HalS-5 | 0.0 | - | HalS-17 | 3.2 | D |
| HalS-6 | 3.0 | L | HalS-18 | 0.0 | - |
| HalS-8 | 2.2 | ++++ | HalS-19 | 0.3 | ++ |
| HalS-9 | 2.4 | ++ | HalS-20 | 3.0 | ++ |
| HalS-10 | 0.0 | - | HalS-21 | 0.0 | +++ |
| HalS-11 | 1.7 | ++++ | HalS-23 | 1.8 | D |
| HalS-12 | 3.2 | D | HalS-24 | 0.5 | ++++ |
| HalS-13 | 0.3 | D | HalS-25 | 1.9 | + |
| Root Nodules of *Vachellia pachyceras* | | | | | |
| Ac-1 | 1.8 | + | LTN-4 | 0.0 | - |
| Ac-2 | 0.0 | - | LTN-5 | 0.0 | - |
| Ac-4 | 0.0 | - | LTN-6 | 0.0 | - |
| Ac-6 | 0.0 | - | LTN-7 | 0.0 | - |
| Ac-7 | 0.0 | - | LTN-9 | 0.0 | - |
| Ac-9 | 1.6 | + | LTN-10 | 2.7 | +++ |
| Ac-10 | 1.6 | + | LTN-11 | 0.0 | - |
| Ac-11 | 0.0 | - | LTN-12 | 0.0 | - |
| Ac-12 | 2.0 | + | LTN-13 | 0.0 | - |
| Ac-13 | 0.0 | - | LTN-14 | 3.4 | ++ |
| Ac-15 | 0.0 | - | LTN-15 | 0.0 | - |
| Ac-16 | 0.0 | - | LTN-17 | 0.0 | - |
| Ac-17 | 0.0 | - | LTN-20 | 0.0 | - |
| Ac-18 | 0.0 | - | ACN-1 | 1.9 | ++ |
| Ac-20 | 0.0 | - | ACN-2 | 2.8 | ++++ |
| Ac-21 | 0.0 | - | ACN-3 | 1.9 | ++++ |
| Ac-22 | 2.1 | + | ACN-4 | 3.6 | L |
| Ac-25 | 1.2 | ++ | ACN-5 | 3.6 | L |
| Ac-26 | 0.0 | - | ACN-6 | 0.0 | L |

*(Continued)*

**Table 1.** (Continued)

| Isolate ID Number | Diameter of Zone of Colorization (cm) | Relative Nitrogen Fixation Potential (Acetylene Reduction Assay) | Isolate ID Number | Diameter of Zone of Colorization (cm) | Relative Nitrogen Fixation Potential (Acetylene Reduction Assay) |
|---|---|---|---|---|---|
| Ac-30 | 2.1 | + | ACN-8 | 0.0 | - |
| Ac-31 | 0.0 | - | ACN-9 | 3.0 | ++ |
| Ac-33 | 0.0 | - | ACN-10 | 0.0 | ++ |
| Ac-34 | 0.0 | - | ACN-11 | 0.0 | - |
| Ac-35 | 2.1 | + | ACN-12 | 0.1 | +++ |
| Ac-36 | 0.0 | - | ACN-13 | 0.0 | - |
| Ac-38 | 0.0 | - | ACN-14 | 0.2 | D |
| Ac-39 | 0.1 | ++++ | ACN-14 (B) | 0.1 | D |
| Ac-40 | 2.1 | + | ACN-15 | 0.0 | - |
| LTN-1 | 0.8 | ++ | ACN-16 | 0.0 | - |
| LTN-2 | 0.0 | - | ACN-17 | 0.1 | D |
| LTN-3 | 0.0 | - | | | |

-: no activity; +: low maximum activity (below 20 nmol/h/OD; ++: moderate maximum activity (20–50 nmol/h/OD); +++: high activity (50–80 nmol/h/OD); ++++: very high maximum activity (above 80 nmol/h/OD); L: Bacteria lost due to contamination; D: Did not grow in N₂ free media to conduct Acetylene Reduction; Rha S: Isolates from the rhizospheric soil of *Rhanterium epapposum*; Far S: Isolates from the rhizospheric soil of *Farsetia aegyptia*; Hal S: Isolates from the rhizospheric soil of *Haloxylon salicornicum*; Ac, ACN, LTN: Isolates from two set of *Vachellia pachyceras* root nodules from KSRI); ARA acetylene reduction assay.

*anthian/caribensis*, *E. homaechei*, and *Microbacterium* spp. (CDC.A-4). Out of 50 isolates tested for biochemical analysis for possible identification, about 23 isolates could not be identified and were marked as "NO ID" (Table 2, Fig 5A). The reason for not being able to identify these isolates could be that thay are new strains or that not all isolates are supported by the current Biolog® database.

Thus, we decided to confirm the identities of isolated strains using molecular identification system using the 16S rRNA gene sequencing method. Partial 16S rRNA gene sequences were obtained for all 50 bacterial strains isolated from the rhizosphere soil samples of *R. epapposum*, *F. aegyptia*, *H. salicornicum* and root nodules of *V. pachyceras*. All the isolates including the strains identified with Biolog® GEN III Microbial Identification System, were identified up to

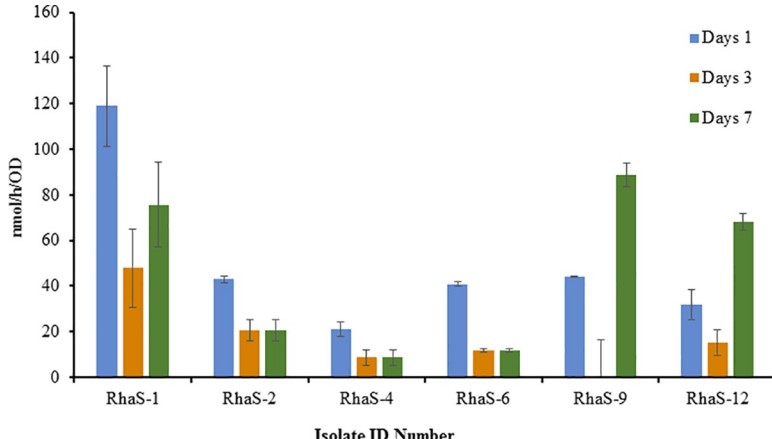

**Fig 1. Ethylene production rate (nmol/h) normalized to cell density (OD) measured in the isolate isolated from the rhizospheric soil of *Rhanterium epapposum* after 1, 3, and 7 days of incubation at 28˚C in solid medium (modified Fraquil).**

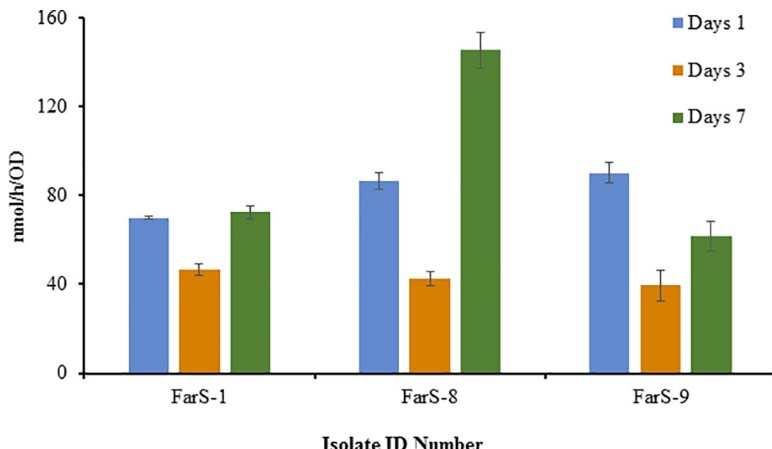

**Fig 2. Ethylene production rate (nmol/h) normalized to cell density (OD) measured in the isolate isolated from the rhizospheric soil of *Farsetia aegyptia* after 1, 3, and 7 days of incubation at 28˚C in solid medium (modified Fraquil).**

the species level using 16S rRNA sequencing method (Table 2). The BLASTn results from 16S rRNA sequences identified a diverse group of bacteria and a majority of the sequences had more than 99% similarity with closely matching sequences existing in the database of Bacterial 16S Ribosomal RNA RefSeq Targeted Loci Project. Only four isolates exhibited less than 99% similarity, *Pseudoxanthomonas japonensis* strain NBRC 101033 (98%), *Pseudomonas glareae* strain KMM 9500 (98%) similarity; *Sphingomonas zeicaulis* strain 541 (97%). Twelve genera were identified among the 50 identified bacterial strains with the dominant species *Rhizobium* (28%) followed by *Pseudomonas* (24%), *Agrobacterium* (18%), *Bacillus* (6%), *Cellulomonas* (6%), *Klebsiella* (4%), *Microbacterium* (4%) *Sphingomonas* (2%), *Arthrobacter* (2%), *Enterobacter* (2%), *Leifsonia* (2%), *Massilia* (2%) (Fig 5B).

Phylogenetic analysis was conducted for representative bacterial species. The ML and Bayesian analysis showed five major groups of bacteria belonging to the class: γ -proteobacteria, α –protoebacteria, and β -proteobacteria; phyla: Actinobacteria, and Firmicutes (Fig 6). γ -proteobacteria was the most dominant group and clustered with seven bacterial species: *P. glareae* strain KMM 9500, *P. koreensis* strain Ps 9–14, *Pseudomonas songnenensis* strain NEAU-ST5-5, *Pseudomonas stutzeri* strain VKM B-975, *Klebsiella pneumoniae* subsp.

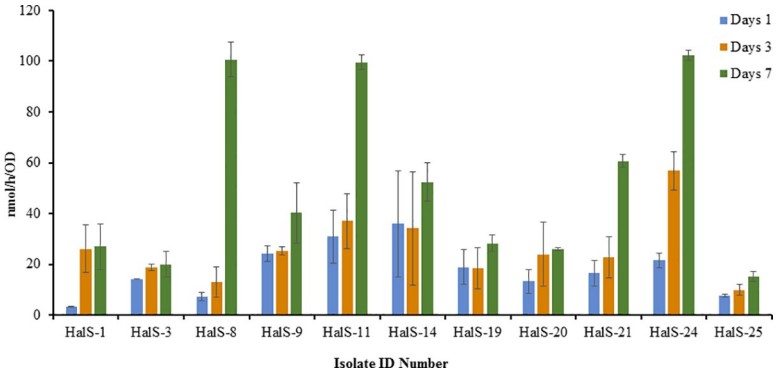

**Fig 3. Ethylene production rate (nmol/h) normalized to cell density (OD) measured in the isolate isolated from the rhizospheric soil of *Haloxylon salicornicum* after 1, 3, and 7 days of incubation at 28˚C in solid medium (modified Fraquil).**

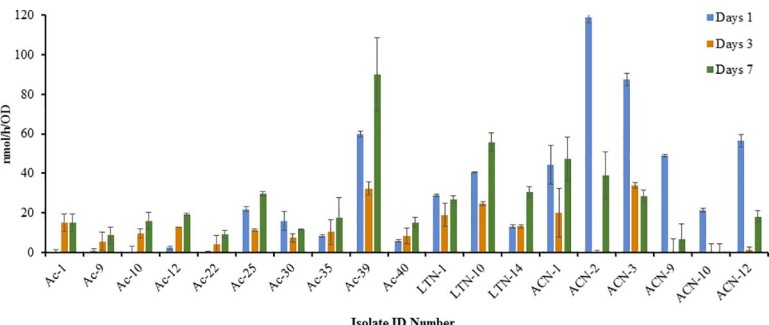

**Fig 4. Ethylene production rate (nmol/h) normalized to cell density (OD) measured in the isolate isolated from the root nodule of *Vachellia pachyceras* after 1, 3, and 7 days of incubation at 28˚C in solid medium (modified Fraquil).**

*rhinoscleromatis*, *Enterobacter cloacae* strain ATCC 13047, and *P. japonensis* strain NBRC 101033. α -proteobacteria and Actinobacteria were the second dominant group and clustered with five and four bacterial species, respectively in each group. In α -proteobacteria, species belonged to *Agrobacterium tumefaciens* strain IAM 12048, *Rhizobium pakistanense* strain BN-19, *Rhizobium subbaraonis* strain JC85, and *S. zeicaulis* strain 541; Actinobacteria, species belonged to *Cellulomonas massiliensis* strain JC225, *Microbacterium assamensis* strain S2-48, *Arthrobacter nitroguajacolicus* strain G2-1, and *Leifsonia shinshuensis* strain DB 102. Two species clustered in the phylum Firmicutes and a single species was present in the class β -proteobacteria. The present study revealed a considerable number of both free-living and root-associated bacterial strains that are present in the desert soil of Kuwait. For the phylogenetic analyses, to identify and position unknown sequences, a different branching pattern in the phylogenetic tree was observed for ML and Bayesian analyses at the phylum level and so did not affect the interpretation; the Bayesian topology is presented in the supplementary material (S1 Fig).

## Discussion

The aim of this investigation was to uncover the diazotrophic bacterial community structure associated with the rhizosphere soils of native keystone shrubs and root nodules of a tree species, and to assess their N$_2$-fixation ability. To our knowledge, this is the first time that culture-based isolation, identification, and evaluation of N$_2$-fixing bacteria from rhizosphere soils has been done for native Kuwaiti plant species to report on the N$_2$-fixing ability of isolated bacterial strains using ARA. Available literatures indicate that most of the studies related to N$_2$-

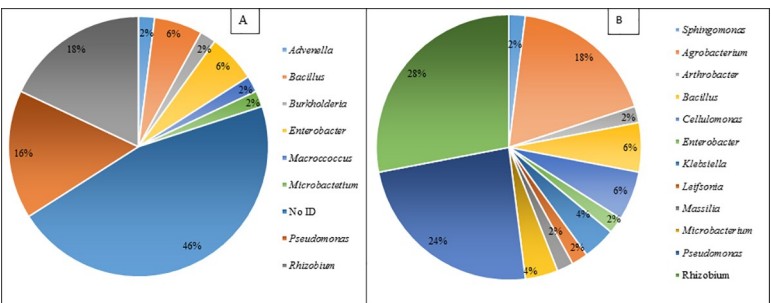

**Fig 5.** Dominant bacterial genera identified in this study (A), biochemical identification using BIOLOG® Gen III Microbial Identification System (B), 16S rRNA sequencing.

**Table 2. Biochemical and molecular identification of the isolates positive for primary screening for nitrogen fixation.**

| Isolate ID Number | Biochemical identification (Biolog Gen III Microbial Identification System) | Molecular Identification System | | |
|---|---|---|---|---|
| | | Isolate Identity | % Identity | NCBI Accession Number |
| RhaS-1 | No ID | *Sphingomonas zeicaulis* strain 541 | 503/518 (97%) | NR_152012 |
| RhaS-2 | *Enterobacter cowanii* | *Klebsiella pneumoniae* subsp. Rhinoscleromatis | 536/543 (99%) | NR_114507 |
| RhaS-4 | No ID | *Rhizobium pakistanense* strain BN-19 | 515/517 (99%) | NR_145565 |
| RhaS-6 | *Enterobacter cowanii* | *Klebsiella pneumoniae* subsp. rhinoscleromatis | 537/543 (99%) | NR_114507 |
| RhaS-9 | No ID | *Rhizobium pakistanense* strain BN-19 | 514/517 (99%) | NR_145565 |
| RhaS-12 | No ID | *Rhizobium pakistanense* strain BN-19 | 515/517 (99%) | NR_145565 |
| FarS-1 | No ID | *Rhizobium pakistanense* strain BN-19 | 501/502 (99%) | NR_145565 |
| FarS-2 | *Advenella incenata* | *Pseudomonas glareae* strain KMM 9500 | 530/537 (99%) | NR_145562 |
| FarS-3 | *Pseudomonas viridilivida* | *Pseudomonas glareae* strain KMM 9500 | 530/537 (99%) | NR_145562 |
| FarS-4 | *Pseudomonas stutzeri* | *Pseudomonas glareae* strain KMM 9500 | 529/539 (98%) | NR_145562 |
| FarS-6 | No ID | *Pseudoxanthomonas japonensis* strain NBRC 101033 | 528/538 (98%) | NR_113972 |
| FarS-7 | No ID | *Pseudoxanthomonas japonensis* strain NBRC 101033 | 528/538 (98%) | NR_113972 |
| FarS-8 | *Pseudomonas stutzeri* | *Pseudomonas stutzeri* strain VKM B-975 | 536/537 (99%) | NR_116489 |
| FarS-9 | No ID | *Rhizobium pakistanense* strain BN-19 | 508/511 (99%) | NR_145565 |
| HalS-1 | No ID | *Rhizobium subbaraonis* strain JC85 | 512/517 (99%) | NR_108508 |
| HalS-3 | No ID | *Rhizobium subbaraonis* strain JC85 | 512/517 (99%) | NR_108508 |
| HalS-8 | *Pseudomonas stutzeri* | *Pseudomonas stutzeri* strain VKM B-975 | 570/571 (99%) | NR_116489 |
| HalS- 9 | No ID | *Rhizobium subbaraonis* strain JC85 | 535/540 (99%) | NR_108508 |
| HalS-11 | No ID | *Rhizobium subbaraonis* strain JC85 | 535/540 (99%) | NR_108508 |
| HalS-12 | *Pseudomonas stutzeri* | *Pseudomonas songnenensis* strain NEAU-ST5-5 | 575/578 (99%) | NR_148295 |
| HalS-13 | No ID | *Rhizobium pakistanense* strain BN-19 | 535/540 (99%) | NR_145565 |
| HalS-14 | No ID | *Rhizobium subbaraonis* strain JC85 | 536/540 (99%) | NR_108508 |
| HalS-17 | No ID | *Massilia timonae* strain UR/MT95 | 562/565 (99%) | NR_026014 |
| HalS-19 | No ID | *Rhizobium pakistanense* strain BN-19 | 536/540 (99%) | NR_145565 |
| HalS-20 | No ID | *Rhizobium subbaraonis* strain JC85 | 535/540 (99%) | NR_108508 |
| HalS-21 | No ID | Leifsonia shinshuensis strain DB 102 | 547/548 (99%) | NR_043663 |

*(Continued)*

**Table 2.** (Continued)

| Isolate ID Number | Biochemical identification (Biolog Gen III Microbial Identification System) | Molecular Identification System | | |
| --- | --- | --- | --- | --- |
| | | Isolate Identity | % Identity | NCBI Accession Number |
| HalS-23 | No ID | *Rhizobium pakistanense* strain BN-19 | 536/540 (99%) | NR_145565 |
| HalS-24 | *Macroccoccus equipercicus* | *Microbacterium assamensis* strain S2-48 | 548/548 (100%) | NR_132711 |
| HalS-25 | No ID | *Microbacterium assamensis* strain S2-48 | 557/558 (99%) | NR_132711 |
| Ac-1 | *Rhizobium radiobacter* | *Agrobacterium tumefaciens* strain IAM 12048 | 540/540 (100%) | NR_041396 |
| Ac-9 | *Rhizobium radiobacter* | *Agrobacterium tumefaciens* strain IAM 12048 | 540/540 (100%) | NR_041396 |
| Ac-10 | *Rhizobium radiobacter* | *Agrobacterium tumefaciens* strain IAM 12048 | 540/540 (100%) | NR_041396 |
| Ac-12 | *Rhizobium radiobacter* | *Agrobacterium tumefaciens* strain IAM 12048 | 540/540 (100%) | NR_041396 |
| Ac-22 | *Rhizobium rhizogenes* | *Agrobacterium tumefaciens* strain IAM 12048 | 539/540 (99%) | NR_041396 |
| Ac-25 | *Rhizobium rhizogenes* | *Agrobacterium tumefaciens* strain IAM 12048 | 538/540 (99%) | NR_041396 |
| Ac-30 | *Rhizobium radiobacter* | *Agrobacterium tumefaciens* strain IAM 12048 | 536/540 (99%) | NR_041396 |
| Ac-35 | *Rhizobium radiobacter* | *Agrobacterium tumefaciens* strain IAM 12048 | 539/540 (99%) | NR_041396 |
| Ac-39 | *Pseudomonas fluorescens* | *Pseudomonas koreensis* strain Ps 9–14 | 530/532 (99%) | NR_025228 |
| Ac-40 | *Rhizobium radiobacter* | *Agrobacterium tumefaciens* strain IAM 12048 | 538/540 (99%) | NR_041396 |
| ACN-1 | *Pseudomonas viridilivida* | *Pseudomonas koreensis* strain Ps 9–14 | 554/556 (99%) | NR_025228 |
| ACN-2 | *Enterobacter homaechei* | *Enterobacter cloacae* strain ATCC 13047 | 574/577 (99%) | NR_118568 |
| ACN-3 | *Pseudomonas viridilivida* | *Pseudomonas koreensis* strain Ps 9–14 | 531/533 (99%) | NR_025228 |
| ACN-9 | *Burkholderia anthian/caribensis* | *Pseudomonas koreensis* strain Ps 9–14 | 536/538 (99%) | NR_025228 |
| ACN-10 | *Bacillus odysseyi* | *Arthrobacter nitroguajacolicus* strain G2-1 | 515/515 (100%) | NR_027199 |
| ACN-12 | No ID | *Cellulomonas massiliensis* strain JC225 | 556/559 (99%) | NR_125601 |
| ACN-14 | *Microbactetium* spp. (CDC.A-4) | *Cellulomonas massiliensis* strain JC225 | 556/559 (99%) | NR_125601 |
| ACN-17 | No ID | *Cellulomonas massiliensis* strain JC225 | 553/558 (99%) | NR_125601 |
| LTN-1 | *Bacillus simplex/butanolivorans* | *Bacillus simplex* strain LMG 11160 | 577/579 (99%) | NR_114919 |
| LTN-10 | No ID | *Bacillus flexus* strain NBRC 15715 | 565/566 (99%) | NR_113800 |
| LTN-14 | *Bacillus megaterium* | *Bacillus flexus* strain NBRC 15715 | 565/567 (99%) | NR_113800 |

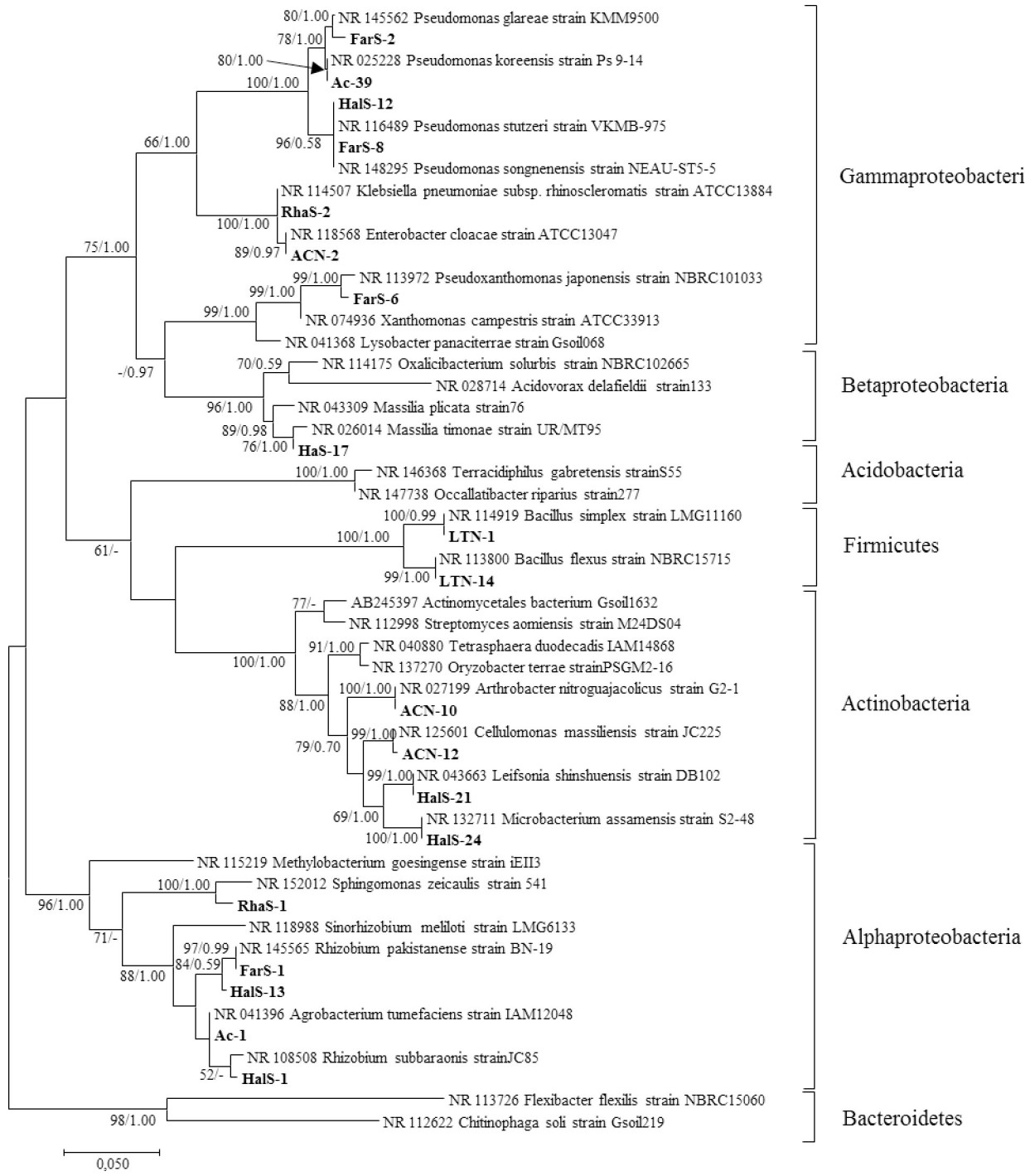

**Fig 6. Maximum likelihood analysis of bacteria.** Bootstrap percentage values (50%) generated from 1000 replicates from maximum likelihood and posterior probabilities (>50%) from Bayesian analysis are shown as [Maximum likelihood bootstrap value/Bayesian posterior probabilities]. Taxa in bold are bacteria from the present study.

fixing bacteria were focused on symbiosis of different leguminous species [37]. There is a lack of information about free-living N$_2$-fixing bacterial community inhabiting desert soils, which perform significant N$_2$-fixation.

## Diversity in free-living nitrogen-fixing bacterial community

Although one-fifth of terrestrial ecosystems comprise arid and semi-arid regions, the diversity and functions of microbial communities are not fully understood [38]. Kuwait desert soils are similar to other desert ecosystems, characterized by extremely harsh environmental conditions with extreme temperatures, high soil salinity, scarce organic matter, low nutrient level, sparse precipitation, high levels of UV radiation, and unstable soil conditions due to dust storms [14, 15, 39]. Despite these prevalence of these conditions in Kuwait, a great diversity of both free-living and root-associated bacterial strains was observed in the present study. Free-living diazotrophs were detected in all rhizosphere soils and roots tested. Approximately 50 isolates (potential nitrogen-fixers) detected from primary screening were further biochemically analyzed; however, only 27 isolates were identified using Biolog® (Table 2). The reason for not identifying all the isolates obtained in primary screening could not be ascertained. We hypothesized that either new strains were present or that not all isolates were present in the Biolog® database. However, all 50 isolates detected from the primary screening were identified using 16S rRNA gene sequencing. Phylogenetic analysis based on 16S rRNA sequences of the representative isolates revealed five groups represented with γ- proteobacteria, α-proteobacteria and Actinobacteria being most dominant and followed by, Firmicutes, and β-proteobacteria (Fig 6). Although the Kuwaiti desert soil is characterized with hostile environmental conditions, our study identified great diversity and relative abundance of nitrogen-fixing bacteria, compared to other ecosystems. There are no previous reports available on the isolation and characterization of free-living or endophytic N$_2$-fixers from the desert soils of Kuwait. Furthermore, the identified bacterial species varied considerably among the rhizosphere soils of different shrubs and root nodules of *V. pachyceras*, suggesting that plant species and their rhizosphere effects are important drivers for specificity of microbial diversity in arid soils. Our report is consistent with several recent reports that acknowledged that plants and root exudates are important drivers for functional gene pool diversity and that specificity of diazotrophic communities is related to the plants species [1, 40–42]. Similarly, a significant difference in free-living nitrogen-fixers was observed based on the type of the plant species, which suggested that these are dominant factors determining the structure of diazotrophs [43]. Recently, Eida [16], evaluated the growth promoting properties and salinity tolerance of the bacterial communities associated with the soil, rhizosphere, and endosphere of four key desert plant species in Saudi Arabia. The study also revealed a dominance of Actinobacteria and Proteobacteria in all samples tested. Although recruitment of microbial communities by plants depends on a number of factors, such as the genotype of the host plant [44, 45], the crucial determinant was recognized as the soil type and the genotype was regarded as a secondary factor in determining microbial composition [46, 47]. In this study, we observed various diazotrophic bacteria related to plant species and supports the recent findings [1, 43].

In general, bacterial communities in most desert soils characteristically consist of a number of ubiquitous phyla including Actinobacteria, Bacteroides, and Proteobacteria [48, 49]. Metagenomic analysis of our current research revealed Proteobacteria and Actinobacteria being the most dominant group, followed by Firmicutes, indicating the typical characteristics of bacterial communities in extreme environmental conditions, such as desert soils. In a recent study, Ren [38] analyzed the microbial community structure of the desert soils from Tarin basin, northwestern China, one of the largest arid regions in China, where water is scarce the year-round. Metagenomic analysis of their study divulged that Proteobacteria, Firmicutes, Actinobacteria, and Euryarchaeota were the most abundant phyla. Proteobacteria are generally presumed to be prominent members of desert soils bacterial communities and may be functionally essential in nutrient-limited arid soils. Another study of free-living nitrogen-fixing microbial

communities in the wastelands of mine tailings, collected from the Tongling copper mine area, East China, reported the presence of α-proteobacteria and β-protobacteria followed by Cyanobacteria, using the nested PCR method [43]. The diversity of the nitrogen-fixers in wasteland varied considerably with the physico-chemical properties of samples, plant species, and rhizosphere [43]. Results from the present study shows that, the isolated bacterial species varied among the rhizosphere soils of different shrubs and root nodules of the test tree species. A recent study conducted by Dahal [50] characterized the diversity of free-living diazotrophs in arid lands of South Dakota, Badlands National Park, USA. Similar to our results, the 16S rRNA sequence data revealed great diversity of putative free-living diazotrophs, belonging to Actinobacteria, Proteobacteria, Bacteroides, and Firmicutes. In contrast, about 50% of these isolates clustered under *Streptomyces*, the first report of nitrogen-fixing by *Streptomyces*. Although our study identified a great diversity in the putative nitrogen-fixing bacterial community, compared to other ecosystems, we suggest that further sampling from across the Kuwaiti landscape may uncover ecologically important and novel nitrogen-fixers.

Diazotrophic bacterial communities and their nitrogen fixation abilities have been investigated in different terrestrial ecosystems and reported in many publications [6]. In contrast to other ecosystems, tropical forest soils are highly weathered and presumed to be phosphorus limited and nitrogen saturated [51], with relatively high levels of nitrogen loss [52]. It is assumed that nitrogen loss in tropical soils may be balanced by high levels of nitrogen fixation [53, 54]. Previously, it was believed that free-living diazotrophs were the dominant type of nitrogen-fixers in temperate forest soils and symbiotic nitrogen-fixers were the dominant type in tropical soils [51]. Results from rain forests of Costa Rica showed that the soils were dominated by *Heliobacterium* a member of Firmicutes and members from α-proteobacteria related to *Azospirillum*, *Gluconaacetobacter*, *Methylobacterium*, and *Zymomonas* [55]. Diazotrophic communities reported from the soils of the western Amazon basin of Brazil showed that the dominant members were α and β Proteobacteria, Firmicutes, and Cyanobacteria [56]. The study also found methanogenic Archaea in these soils and suggested that archaeal biological N-fixation could perform an essential role in the Amazon rainforest [56].

## Assessment of nitrogen-fixing ability by acetylene reduction assay (ARA)

In the present study, among the isolated bacterial strains tested positive from primary screening, about 11 isolates failed to grow in actual nitrogen free media used for ARA. It was difficult to predict why some of these strains did not grow on the modified Fraquil medium. It is likely that, some cultures were affected during the storage period, suggesting repetition of this procedure with fresh cultures in future studies. The experience in isolating N$_2$-fixers from the environment suggests that the N$_2$-free selective medium used should be maintained in N$_2$-free conditions, any contamination may lead to misleading conclusions and interrupts ARA, which happened in this case, and the isolates were re-grown in N$_2$-free modified Fraquil medium for conducting successful ARA. Our study successfully identified all the 50 nitrogen fixers isolated initially using 16S rRNA gene sequencing, and 78% of these were confirmed to be nitrogen fixers using ARA. Among them, most of the species belonged to the free-living *Rhizobium*, *Pseudomonas*, and *Agrobacterium* genera followed by *Cellulomonas* and *Bacillus*. A great diversity of diazotrophs was detected in all samples tested from the desert environment, indicating the immense importance of free-living and root-associated bacteria in harsh conditions and suggesting a great ecological importance of diazotrophs in the desert ecosystem

The rates of conversion of acetylene (C$_2$H$_2$) to ethylene (C$_2$H$_4$) by different bacterial strains tested in this study are lower compared to that of established isolates. However, the results are comparable and similar to the studies conducted by Gothwal [24] and Kifle [57] on the

diazotrophic bacteria isolated from rhizosphere soils of some important desert plants and maize seedlings. Xu [58] examined putative N$_2$-fixing bacteria isolated from rhizosphere soil, root, and stem samples from switchgrass and giant reed and tested for their N$_2$-fixing ability, intended to be used as potential biofertilizers. In this study, the levels of ethylene production in ARA ranged among the strains from 40 to 350 nmol C$_2$H$_4$ 24 h$^{-1}$ mL$^{-1}$, which are comparable to our results. Moreover, acetylene reduction in the ARA test and nitrogenase activity level may also depend on several environmentally and genetically induced factors, such as duration of incubation period and characteristics of the strains. Interestingly, most of the isolated rhizobacterial strains (except Ac 39) from root nodules of *V. pachyceras* (Ac) showed a low maximum level of nitrogen fixation potential, through acetylene reduction assays, compared to all the other strains isolated for this study. The reason for the observed low level of N$_2$-fixation potential could not be confirmed under the scope of this investigation. However, the bacterial strains associated with the root nodules of *V. pachyceras* (Ac) identified as *A. tumefaciens*, which may not be viewed as well known diazotrophs.

## Root nodules like structure of *V. pachyceras* and *A. tumefaciens*

An unexpected observation during the identification of isolated bacterial strains (Table 2), using molecular techniques was made. The rhizobacterial strains isolated from root nodules of *V. pachyceras* (Ac) from Sulaibiya, which were initially identified as *R. radiobacter* by the Biolog GEN III Microbial Identification System, and recorded as positive for N$_2$-fixation initially by the plate test was molecularly identified as *A. tumefaciens* (Table 2). Several reports have indicated that *A. tumefaciens* is a synonym of *R. radiobacter* [59]. The current characterization of the bacterial isolates of this study is mainly based on the partial 16S rRNA gene sequences, which may limit the full exploration of related bacterial species. We suggest that more studies are required to fully characterize bacterial strains based on the *nod*D and *nif*H gene sequences. The endosymbionts in root nodules of legumes possesses *nod* and *nif* genes responsible for nodule development and atmospheric nitrogen fixation, respectively [60]. It is known that *Rhizobium* strains produce root nodules and some strains induce tumors or hairy roots in different species [61]. Nevertheless, this is a noteworthy observation, based on 16S rRNA sequence of strains of *A. tumefaciens*, in which high similarity (99% - 100%) with the *R. radiobacter* strain detected, confirms that both are probably closely related. In this study, we observed evidence of N$_2$-fixing ability of *A. tumefaciens which* is interesting and of practical importance. However, further investigation is necessary to confirm if *A. tumefaciens* may also have the ability to fix atmospheric nitrogen, under desert conditions. The observation from this study also indicates that biochemical test (Biolog®) for microbial identification may not be the most reliable test for identification.

## *A. tumefaciens* as diazotrophs

Although rhizobacterial strains isolated from the root-nodule like structure of *V. pachyceras* (Ac), were identified initially as *R. radiobacter* by biochemical tests, the isolates were eventually identified to be *A. tumefaciens* using 16S rRNA analysis. *A. tumefaciens* is a gram-negative bacterium belonging to the family Rhizobiaceae and is known to cause crown gall disease in the roots of many naturally occurring plants.The bacterium is closely related to the genus *Rhizobium* and belongs to the same family [62]. *A. tumefaciens* can live freely in soils, root surface, and inside the plants as a parasite. In this investigation, a structure resembling a root nodule, often found in the roots of Leguminaceae, was observed and was similar to nitrogen-fixing root nodules produced by *Rhizobium* sp. (S2 Fig). A few studies reported that although *A. tumefaciens* is known as a pathogenic bacterium that cause for crown gall formation in roots,

it was also found to be similar to diazotrophs and can fix nitrogen to grow on nitrogen-free medium [62, 63]. However, we report for the first time, to our knowledge, that *A. tumefaciens* can produce a root nodule type structure with *V. pachyceras* roots, grow on nitrogen-free medium, and reduce acetylene to ethylene at low levels. The isolated strain acts as an associative diazotroph, as suggested by My [63]. The current study recommends further investigation and full understanding of root nodule formation ability and nitrogen-fixating capacity of this bacterium, isolated from both rhizosphere soils and root structure. Further research can bring some understanding on how the pathogenic bacterium transforms to having symbiotic nitrogen-fixing relationships with the host.

## Conclusion

We successfully isolated strains of free-living $N_2$-fixing bacteria from rhizosphere soil samples of three important native shrubs and $N_2$-fixers from root nodules of native plant species, *V. pachyceras*. We identified 50 free-living $N_2$-fixing bacteriain all rhizosphere soil samples tested initially using 16S rRNA gene sequencing, out of which 78% were confirmed to be nitrogen-fixers, using ARA. Among them, most species belonged to the Rhizobium (28%), Pseudomonas (24%), and Agrobacterium (18%) genera followed by Cellulomonas (6%), Bacillus (6%) Klebsiella (4%), Microbacterium (4%), Sphingomonas (2%) Arthrobacter (2%), Enterobacter (2%), Leifsonia (2%), and Massilia (2%). Our results support immense role of these diazotrophs for plant-available nitrogen with native plants in desert ecosystem. Further research needs to be conducted on the identified isolates to evaluate their nitrogen-fixing ability in an arid environment. The current study also warrants additional in-depth research using nodD and nifH gene sequences with appropriate primer sets to confirm nitrogen-fixating capacity of *A. tumefaciens* isolated from the root structure of the desert inhabiting *V. pachyceras*. Inoculation of isolated diazotrophs with different beneficial strains might be the potential trend of using bio fertilizer application for sustainable native plant restoration and revegetation.

## Supporting information

**S1 Fig. Bayesian topology of bacterial isolates supporting the branching pattern of identified bacterial isolates.**
(JPG)

**S2 Fig. The root nodule structure.**
(JPG)

## Acknowledgments

The authors are thankful for the constant cooperation and support provided by the management, other departments of KISR and also the assistance provided by the laboratory and field helpers with this research. The authors also acknowledge the technical assistance from the molecular research team at the Université Laval, Québec, Canada for the molecular identification of bacterial strains and Département de chimie, Université de Sherbrooke for acetylene reduction assay.

## Author Contributions

**Conceptualization:** M. K. Suleiman, A. M. Quoreshi.

**Formal analysis:** A. J. Manuvel, M. T. Sivadasan.

**Funding acquisition:** M. K. Suleiman, A. M. Quoreshi, N. R. Bhat.

**Investigation:** M. K. Suleiman, A. M. Quoreshi, N. R. Bhat, A. J. Manuvel, M. T. Sivadasan.

**Methodology:** M. K. Suleiman, A. M. Quoreshi, N. R. Bhat, A. J. Manuvel, M. T. Sivadasan.

**Project administration:** M. K. Suleiman.

**Supervision:** M. K. Suleiman, A. M. Quoreshi.

**Validation:** M. K. Suleiman, A. M. Quoreshi.

**Writing – original draft:** M. K. Suleiman, A. M. Quoreshi, A. J. Manuvel.

**Writing – review & editing:** M. K. Suleiman, A. M. Quoreshi, A. J. Manuvel.

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
