## [Decision Letter · Decision Letter 0]

25 Sep 2019

PONE-D-19-20527

Divulging diazotrophic bacterial community structure in Kuwait desert ecosystems and their N2-fixation potential

PLOS ONE

Dear Dr. Quoreshi,

Thank you for submitting your manuscript to PLOS ONE. After careful consideration, we feel that it has merit but does not fully meet PLOS ONE’s publication criteria as it currently stands. Therefore, we invite you to submit a revised version of the manuscript that addresses the points raised during the review process.

We would appreciate receiving your revised manuscript by Nov 09 2019 11:59PM. To enhance the reproducibility of your results, we recommend that if applicable you deposit your laboratory protocols in protocols.io, where a protocol can be assigned its own identifier (DOI) such that it can be cited independently in the future. For instructions see: http://journals.plos.org/plosone/s/submission-guidelines#loc-laboratory-protocols

We look forward to receiving your revised manuscript.

Kind regards,

Andrew R. Zimmerman, PhD

Academic Editor

PLOS ONE

Journal Requirements:

1. In your Methods section, please provide additional information regarding the permits you obtained for the work. Please ensure you have included the full name of the authority that approved the field site access and, if no permits were required, a brief statement explaining why.

Additional Editor Comments (if provided):

The manuscript is suitable for publication but needs some minor adjustments. please make the changes indicated by the 2 external reviewers. In addition, I ask that you try to have the text edited by a native English speaker as there are many grammatical errors. In addition, a reviewer commented that "more logical explanations which would definitely enhance the quality of this manuscript". Thus, I ask that you go through the Discussion section, making sure the argument flows logically. Perhaps add some subheadings to indicate topics being discussed and divide some of the longer paragraphs into a few shorter ones.

Reviewers' comments:

Reviewer's Responses to Questions

**Comments to the Author**

1. Is the manuscript technically sound, and do the data support the conclusions?

Reviewer #1: Yes

Reviewer #2: Yes

2. Has the statistical analysis been performed appropriately and rigorously? 

Reviewer #1: Yes

Reviewer #2: N/A

3. Have the authors made all data underlying the findings in their manuscript fully available?

Reviewer #1: No

Reviewer #2: Yes

4. Is the manuscript presented in an intelligible fashion and written in standard English?

Reviewer #1: Yes

Reviewer #2: Yes

5. Review Comments to the Author

Reviewer #1: 1. At line no. 27 you have used BNF as abbreviation for Biological nitrogen fixation. However, at line no 74 you are using same abbreviation for Biological nitrogen fixers. Please correct this, both biological nitrogen fixation and biological nitrogen fixers are absolutely different.

2. At line no 116, please explain the methodology of rhizospheric soil sample collection in details.

3. Please put space between number and °C throughout the manuscript. For example write 10 °C instead of 10°C.

4. Please make the tables compact. It would be easy to understand data if more information is present in single page rather than the multiple pages.

5. In line no 434 please replace N2 with N2.

6. At line no 461 the authors have written “In this investigation, a structure resembles to root nodule often found in Leguminaceae plant roots was observed and assumed similar to typical nitrogen-fixing root nodules produced by Rhizobium sp.”. How could you make this statement without showing results of nodule characterization? Please justify the about statement written in the manuscript.

7. Image quality of figures is very poor. Please incorporate high quality images.

8. This study is showing that Agrobacterium tumefaciens, which has been renamed as Rhizobium radiobacter, was recovered by root nodules and have potential to fix atmospheric nitrogen outside of the host as shown in ARA based analysis. However, other nodules forming Rhizobium species does not fix nitrogen outside of the host. Moreover, characterization of the bacterial isolates is only based on the partial 16S rDNA sequencing which do not have sufficient resolution to characterize closely related bacterial species. Therefore, please explain your results more clearly and logically. Please go through the following article as it can help you to explain your results in more details:

Velázquez, E., Peix, A., Zurdo-Piñiro, J. L., Palomo, J. L., Mateos, P. F., Rivas, R., ... & Martínez-Molina, E. (2005). The coexistence of symbiosis and pathogenicity-determining genes in Rhizobium rhizogenes strains enables them to induce nodules and tumors or hairy roots in plants. Molecular plant-microbe interactions, 18(12), 1325-1332.

Reviewer #2: The authors enriched and isolated several number of free and symbiotic N2-fixers from rhizosphere and root nodes of four plant species in Kuwait semi-arid desert ecosystem. Apart from precipitation, nitrogen is important for plant growth and microbial nitrogen-fixers play essential roles in nitrogen supply for the semi-arid desert ecosystem. The isolation of many nitrogen-fixing bacteria can pave the way to fully understand N cycling in the special ecosystem, and it shows the most significance especially in the modern popular "metagenomics" era. The manuscript has been well organized and the materials and methods were well described. There was only minor parts needed to be revised: Discuss the differences in microbial compositions between this study and other studies/ ecosystems.

L34-36, repetead sentences "In this study,......species"

L84, precipitation is also important

L375, references?

L376, Are there any differences in microbial compositions between this study and other studies/ ecosystems? What's the difference?

6. PLOS authors have the option to publish the peer review history of their article (what does this mean?). If published, this will include your full peer review and any attached files.

Reviewer #1: No

Reviewer #2: No

---

## [Author Response · Author response to Decision Letter 0]

29 Oct 2019

Editor Comments:

C1: Journal Requirements:

When submitting your revision, we need you to address these additional requirements. Please ensure that your manuscript meets PLOS ONE's style requirements, including those for file naming. The PLOS ONE style templates can be found at

R1: We followed the PLOS ONE author instructions to meet PLOS ONE’s style requirement as advised. 

C2: In your Methods section, please provide additional information regarding the permits you obtained for the work. Please ensure you have included the full name of the authority that approved the field site access and, if no permits were required, a brief statement explaining why.

R 2: We understand the reviewer’s concern regarding the permits obtained for the work. However, all the samples for this study were collected from KISR’s Station for Research and Innovation (KSRI). KSRI is the research station that belongs to Kuwait Institute for Scientific Research (KISR). As per KISR’s rule once the research project is approved by KISR, automatically access is also granted to collect the sample from KSRI for research purpose. Hence, special permit was not obtained for sample collection from KSRI for this study. 

C3: We suggest you thoroughly copyedit your manuscript for language usage, spelling, and grammar. If you do not know anyone who can help you do this, you may wish to consider employing a professional scientific editing service.Whilst you may use any professional scientific editing service of your choice, PLOS has partnered with both American Journal Experts (AJE) and Editage to provide discounted services to PLOS authors. Both organizations have experience helping authors meet PLOS guidelines and can provide language editing, translation, manuscript formatting, and figure formatting to ensure your manuscript meets our submission guidelines. To take advantage of our partnership with AJE, visit the AJE website (http://learn.aje.com/plos/) for a 15% discount off AJE services. To take advantage of our partnership with Editage, visit the Editage website (www.editage.com) and enter referral code PLOSEDIT for a 15% discount off Editage services. If the PLOS editorial team finds any language issues in text that either AJE or Editage has edited, the service provider will re-edit the text for free.

R3: As suggested by the Editor, we submitted our revised manuscript to the Editage services (www.editage.com) for professional editing. The editing service was excellent, fast and professional. We have incorporated all the changes made by the professional editor as well as our responses to the two reviewers’ comments into the manuscript. Now the re-submitted revised manuscript is fully revised and edited for language usage, spelling, and grammar. A certificate provided from Editage Services is attached at the bottom of this document. The original track change document from Editage Services is also uploaded as a supporting information file for your reference. 

C4: We note that you have included the phrase “data not shown” in your manuscript. Unfortunately, this does not meet our data sharing requirements. PLOS does not permit references to inaccessible data. We require that authors provide all relevant data within the paper, Supporting Information files, or in an acceptable, public repository. Please add a citation to support this phrase or upload the data that corresponds with these findings to a stable repository (such as Figshare or Dryad) and provide and URLs, DOIs, or accession numbers that may be used to access these data. Or, if the data are not a core part of the research being presented in your study, we ask that you remove the phrase that refers to these data. 

R4: We sincerely apologize for including the phrase “data not shown” in the manuscript. As the data for the Acetylene Reduction Assay (ARA) conducted using Yeast Manitol media is not a core part of the research, it is removed from the manuscript. The results of Acetylene Reduction Assay conducted using nitrogen free media (modified Fraquil medium) is the important core part of the research and hence the complete data related to that experiment is presented in the manuscript.

Additional Editor Comments (if provided):

C5: The manuscript is suitable for publication but needs some minor adjustments. Please make the changes indicated by the 2 external reviewers. In addition, I ask that you try to have the text edited by a native English speaker, as there are many grammatical errors. In addition, a reviewer commented that "more logical explanations which would definitely enhance the quality of this manuscript". Thus, I ask that you go through the Discussion section, making sure the argument flows logically. Perhaps add some subheadings to indicate topics being discussed and divide some of the longer paragraphs into a few shorter ones.

R5: We appreciate additional Editor Comments on our manuscript. As advised by the Editor, we incorporated all the changes indicated by the two external reviewers and changes are described under each reviewer comments. As mentioned under C3: the revised text is fully edited from professional editing service (www.editage.com). We believe the quality of this manuscript is now enhanced substantially. Furthermore, as suggested by the Editor, we thoroughly re-visited the Discussion section and revised; bring more logical discussion, added new references particularly under the sub-section “Diversity in free-living nitrogen-fixing bacterial community”. As suggested, the whole Discussion section is now divided into four sub-sections which reflected better about the topics being discussed and also divided some of the longer paragraphs into shorter ones as suggested. 

Review Comments to the Author:

Reviewer #1: 

C1: At line no. 27 you have used BNF as abbreviation for Biological nitrogen fixation. However, at line no 74 you are using same abbreviation for Biological nitrogen fixers. Please correct this, both biological nitrogen fixation and biological nitrogen fixers are absolutely different.

R1: Indeed, BNF is the abbreviation for Biological Nitrogen Fixation. Therefore, the BNF in line no. 74 was corrected to ‘biological nitrogen fixers’. We deeply apologize for the confusion. 

C2: At line no 116, please explain the methodology of rhizospheric soil sample collection in details.

R2: As per reviewer’s advice, the methodology for soil and root nodule sample collection was revised completely in detail and incorporated under the section “Sampling” in the manuscript. 

C3: Please put space between number and °C throughout the manuscript. For example write 10 °C instead of 10°C.

R3: As per the reviewer’s advice, a space was added between the number and °C throughout the manuscript.

C4: Please make the tables compact. It would be easy to understand data if more information is present in single page rather than the multiple pages.

R4: We agree with the reviewer that it is easy to understand the data if more information is present in a single page rather than the multiple pages. We tried our best to compress the table as suggested. Consequently, the two tables were compressed in to four pages from the original eight pages and are presented in the manuscript.

C5: In line no 434 please replace N2 with N2.

R5: We apologize for the mistake in line no. 434. The N2 in line no. 434 was replaced with N2. Also ‘N2’ was checked throughout the manuscript and corrected.

C6: At line no 461 the authors have written “In this investigation, a structure resembles to root nodule often found in Leguminaceae plant roots was observed and assumed similar to typical nitrogen-fixing root nodules produced by Rhizobium sp.”. How could you make this statement without showing results of nodule characterization? Please justify the about statement written in the manuscript.

R6: We intensely appreciate the reviewer’s concern on nodule characterization. In order to justify the statement “In this investigation, a structure resembles to root nodule often found in Leguminaceae plant roots was observed and assumed similar to typical nitrogen-fixing root nodules produced by Rhizobium sp.”, the picture of root nodule documented during the root nodule sample collection is added in the supplementary material ( S2 Fig) for reference. 

C7: Image quality of figures is very poor. Please incorporate high quality images.

R7: As per the reviewer’s advice, high quality of the figures with 300dpi resolution were produced and incorporated in the manuscript during the resubmission. We hope the current figures are better quality. 

C8: This study is showing that Agrobacterium tumefaciens, which has been renamed as Rhizobium radiobacter, was recovered by root nodules and have potential to fix atmospheric nitrogen outside of the host as shown in ARA based analysis. However, other nodules forming Rhizobium species does not fix nitrogen outside of the host. Moreover, characterization of the bacterial isolates is only based on the partial 16S rDNA sequencing which do not have sufficient resolution to characterize closely related bacterial species. Therefore, please explain your results more clearly and logically. Please go through the following article as it can help you to explain your results in more details:

Velázquez, E., Peix, A., Zurdo-Piñiro, J. L., Palomo, J. L., Mateos, P. F., Rivas, R., ... & Martínez-Molina, E. (2005). The coexistence of symbiosis and pathogenicity-determining genes in Rhizobium rhizogenes strains enables them to induce nodules and tumors or hairy roots in plants. Molecular plant-microbe interactions, 18(12), 1325-1332.

R8: We appreciate reviewer comments on the above concern. We re-visited the discussion section and tried to discuss our results more elaborately and therefore a completely new discussion is added with few new references, particularly under sub-sections: Diversity in free-living nitrogen-fixing bacterial community and Root nodules like structure of V. pachyceras. In order to discuss more clearly, the discussion section is now divided into four sub-sections, such as 

1. Diversity in free-living nitrogen-fixing bacterial community 

2. Assessment of nitrogen-fixing ability by acetylene reduction assay (ARA)

3. Root nodules like structure of V. pachyceras and A. tumefaciens 

4. A. tumefaciens as diazotrophs

Reviewer #2: 

C1: The authors enriched and isolated several number of free and symbiotic N2-fixers from rhizosphere and root nodes of four plant species in Kuwait semi-arid desert ecosystem. Apart from precipitation, nitrogen is important for plant growth and microbial nitrogen-fixers play essential roles in nitrogen supply for the semi-arid desert ecosystem. The isolation of many nitrogen-fixing bacteria can pave the way to fully understand N cycling in the special ecosystem, and it shows the most significance especially in the modern popular "metagenomics" era. The manuscript has been well organized and the materials and methods were well described. There was only minor parts needed to be revised: Discuss the differences in microbial compositions between this study and other studies/ ecosystems.

R1: We really appreciate reviewer overall comments on our manuscript. We thank reviewer for directing about bringing some discussion on the differences in microbial compositions between this study and other studies elsewhere. As suggested, we added some new discussions in the last two paragraphs under the sub-section “Diversity in free-living nitrogen-fixing bacterial community”. 

C2: L34-36, repetead sentences "In this study,......species"

R2: We sincerely apologize for the repetition of the sentences in L34-36. The repeated sentences were removed and revised in the manuscript.

C3: L84, precipitation is also important

R3: We agree with the comment. As suggested, we revised the sentence and added about the precipitation information with a new reference. 

C4: L375, references?

R4: As suggested by the reviewer, we added new references to support the statement. 

C5: L376, Are there any differences in microbial compositions between this study and other studies/ ecosystems? What's the difference?

R5: Yes, there are differences in microbial compositions between the ecosystems. To address reviewer concern, we added some new discussions in the last two paragraphs under the sub-section “Diversity in free-living nitrogen-fixing bacterial community”.

---

## [Editor Report · Decision Letter 1]

31 Oct 2019

PONE-D-19-20527R1

Divulging diazotrophic bacterial community structure in Kuwait desert ecosystems and their N2-fixation potential

PLOS ONE

Dear Dr. Quoreshi,

Thank you for submitting your manuscript to PLOS ONE. After careful consideration, we feel that it has merit but does not fully meet PLOS ONE’s publication criteria as it currently stands. Therefore, we invite you to submit a revised version of the manuscript that addresses the points raised during the review process.

We would appreciate receiving your revised manuscript by Dec 15 2019 11:59PM. To enhance the reproducibility of your results, we recommend that if applicable you deposit your laboratory protocols in protocols.io, where a protocol can be assigned its own identifier (DOI) such that it can be cited independently in the future. For instructions see: http://journals.plos.org/plosone/s/submission-guidelines#loc-laboratory-protocols

We look forward to receiving your revised manuscript.

Kind regards,

Andrew R. Zimmerman, PhD

Academic Editor

PLOS ONE

Additional Editor Comments (if provided):

Dear Suleiman,

I have just a few more requests of you. First, I have further edited your abstract as I feel this is really important to get right. See if you agree with my suggestions:

(Track change version in the attached) Kuwait is a semi-arid region with soils that are relatively nitrogen-poor. Thus, biological nitrogen fixation is an important natural process in which N2-fixing bacteria (diazotrophs) convert atmospheric nitrogen into plant-usable forms such as ammonium and nitrate. Currently, there is limited information on free-living and root-associated nitrogen-fixing bacteria and their potential to fix nitrogen and aid natural plant communities in the Kuwait desert. In this study, free living N2-fixing diazotrophs were enriched and isolated from the rhizosphere soil associated with three native keystone plant species; Rhanterium epapposum, Farsetia aegyptia, and Haloxylon salicornicum. Root-associated bacteria were isolated from the root nodules of Vachellia pachyceras. The result showed that the strains were clustered in five groups including the Actinobacteria being the most dominant phyla followed by Firmicutes, and the classes β-proteobacteria γ-proteobacteria, and α-proteobacteriaThis study initially identified 50 nitrogen-fixers by16S rRNA gene sequencing, of which 78% were confirmed to be nitrogen-fixers using the acetylene reduction assay. Among the nitrogen fixers identified, the genus Rhizobium was predominant in the rhizosphere soil of R. epapposum and H. salicornicum, whereas Pseudomonas was predominant in the rhizosphere soil of F. aegyptia, . The species Agrobacterium tumefaciens was dominant in the root nodules of V. pachyceras. These results indicate that plant species and their rhizosphere effects are important drivers of diazotroph diversity in arid soils. To our knowledge, this study is the first to use culture-based isolation, molecular identification, and evaluation of N2-fixing ability to detail diazotroph diversity in Kuwaiti desert soils.

Second, I would like you to put a bit more thought into your conclusion section. The statement "most species belonged to the Rhizobium, Pseudomonas, and Agrobacterium genera followed by Cellulomonas, and Bacillus genera" seems quite different from what you have in the abstract. Or maybe I misunderstand and this information should go into the abstract.

But beyond that, the abstract should not just restate the results. I would rather like to see it include a statement of the broader ecological/scientific significance of the results.

---

## [Author Response · Author response to Decision Letter 1]

17 Nov 2019

Response to reviewer’s comments

(PONE-D-19-20527 R1)

C1: I have just a few more requests of you. First, I have further edited your abstract as I feel this is really important to get right. See if you agree with my suggestions:

R1: We immensely appreciate the editor’s suggestions on the abstract of the manuscript. We had accepted most of the changes requested by the editor and incorporated the ecological/scientific significance of the results and the revised abstract is given below. However, we did not agree the changes made on the highlighted sentence, as the order of the bacterial groups were defined according to the number of strains in each group (refer to figure 6).

Abstract:

Kuwait is a semi-arid region with soils that are relatively nitrogen-poor. Thus, biological nitrogen fixation is an important natural process in which N2-fixing bacteria (diazotrophs) convert atmospheric nitrogen into plant-usable forms such as ammonium and nitrate. Currently, there is limited information on free-living and root-associated nitrogen-fixing bacteria and their potential to fix nitrogen and aid natural plant communities in the Kuwait desert. In this study, free living N2-fixing diazotrophs were enriched and isolated from the rhizosphere soil associated with three native keystone plant species; Rhanterium epapposum, Farsetia aegyptia, and Haloxylon salicornicum. Root-associated bacteria were isolated from the root nodules of Vachellia pachyceras. The result showed that the strains were clustered in five groups represented by class: γ-proteobacteria, and α-proteobacteria; phyla: Actinobacteria being the most dominant, followed by phyla: Firmicutes, and class: β-proteobacteria. This study initially identified 50 nitrogen-fixers by16S rRNA gene sequencing, of which 78% were confirmed to be nitrogen-fixers using the acetylene reduction assay. Among the nitrogen fixers identified, the genus Rhizobium was predominant in the rhizosphere soil of R. epapposum and H. salicornicum, whereas Pseudomonas was predominant in the rhizosphere soil of F. aegyptia, The species Agrobacterium tumefaciens was mainly found to be dominant among the root nodules of V. pachyceras and followed by Cellulomonas, Bacillus, and Pseudomonas genera as root-associated bacteria. The variety of diazotrophs revealed in this study, signifying the enormous importance of free-living and root-associated bacteria in extreme conditions and suggesting potential ecological importance of diazotrophs in arid ecosystem. To our knowledge, this study is the first to use culture-based isolation, molecular identification, and evaluation of N2-fixing ability to detail diazotroph diversity in Kuwaiti desert soils.

C2: Second, I would like you to put a bit more thought into your conclusion section. The statement "most species belonged to the Rhizobium, Pseudomonas, and Agrobacterium genera followed by Cellulomonas, and Bacillus genera" seems quite different from what you have in the abstract. Or maybe I misunderstand and this information should go into the abstract.

R2: We agree that the information regarding the major genera were incomplete and it is revised completely and added to the revised manuscript in the conclusion section. We apologize for the misunderstanding.The revised conclusion is given below.

Conclusion:

We successfully isolated strains of free-living N2-fixing bacteria from rhizosphere soil samples of three important native shrubs and N2-fixers from root nodules of native plant species, V. pachyceras. We identified 50 free-living N2-fixing bacteriain all rhizosphere soil samples tested initially using 16S rRNA gene sequencing, out of which 78 % were confirmed to be nitrogen-fixers, using ARA. Among them, most species belonged to the Rhizobium (28 %), Pseudomonas (24 %), and Agrobacterium (18 %) genera followed by Cellulomonas (6 %), Bacillus (6 %) Klebsiella (4 %), Microbacterium (4 %), Sphingomonas (2 %) Arthrobacter (2 %), Enterobacter (2 %), Leifsonia (2 %), and Massilia (2 %). Our results supports immense role of these diazotrophs for plant-available nitrogen with native plants in desert ecosystem. genera. Further research needs to be conducted on the identified isolates to evaluate their nitrogen-fixing ability in an arid environment. The current study also warrants additional in-depth research using nodD and nifH gene sequences with appropriate primer sets to confirm nitrogen-fixating capacity of A. tumefaciens isolated from the root structure of the desert inhabiting V. pachyceras. Inoculation of isolated diazotrophs with different beneficial strains might be the potential trend of using bio fertilizer application for sustainable native plant restoration and revegetation.

C3: But beyond that, the abstract should not just restate the results. I would rather like to see it include a statement of the broader ecological/scientific significance of the results.

R3: As mentioned earlier, the abstract was revised to incorporate the broader ecological/scientific significance of the results rather than just restate the results.

Response to reviewer’s comments

(PONE-D-19-20527)

Editor Comments

C1: Journal Requirements:

When submitting your revision, we need you to address these additional requirements. Please ensure that your manuscript meets PLOS ONE's style requirements, including those for file naming. The PLOS ONE style templates can be found at

R1: We followed the PLOS ONE author instructions to meet PLOS ONE’s style requirement as advised. 

C2: In your Methods section, please provide additional information regarding the permits you obtained for the work. Please ensure you have included the full name of the authority that approved the field site access and, if no permits were required, a brief statement explaining why.

R 2: We understand the reviewer’s concern regarding the permits obtained for the work. However, all the samples for this study were collected from KISR’s Station for Research and Innovation (KSRI). KSRI is the research station that belongs to Kuwait Institute for Scientific Research (KISR). As per KISR’s rule once the research project is approved by KISR, automatically access is also granted to collect the sample from KSRI for research purpose. Hence, special permit was not obtained for sample collection from KSRI for this study. 

C3: We suggest you thoroughly copyedit your manuscript for language usage, spelling, and grammar. If you do not know anyone who can help you do this, you may wish to consider employing a professional scientific editing service.Whilst you may use any professional scientific editing service of your choice, PLOS has partnered with both American Journal Experts (AJE) and Editage to provide discounted services to PLOS authors. Both organizations have experience helping authors meet PLOS guidelines and can provide language editing, translation, manuscript formatting, and figure formatting to ensure your manuscript meets our submission guidelines. To take advantage of our partnership with AJE, visit the AJE website (http://learn.aje.com/plos/) for a 15% discount off AJE services. To take advantage of our partnership with Editage, visit the Editage website (www.editage.com) and enter referral code PLOSEDIT for a 15% discount off Editage services. If the PLOS editorial team finds any language issues in text that either AJE or Editage has edited, the service provider will re-edit the text for free.

R3: As suggested by the Editor, we submitted our revised manuscript to the Editage services (www.editage.com) for professional editing. The editing service was excellent, fast and professional. We have incorporated all the changes made by the professional editor as well as our responses to the two reviewers’ comments into the manuscript. Now the re-submitted revised manuscript is fully revised and edited for language usage, spelling, and grammar. A certificate provided from Editage Services is attached at the bottom of this document. The original track change document from Editage Services is also uploaded as a supporting information file for your reference. 

C4: We note that you have included the phrase “data not shown” in your manuscript. Unfortunately, this does not meet our data sharing requirements. PLOS does not permit references to inaccessible data. We require that authors provide all relevant data within the paper, Supporting Information files, or in an acceptable, public repository. Please add a citation to support this phrase or upload the data that corresponds with these findings to a stable repository (such as Figshare or Dryad) and provide and URLs, DOIs, or accession numbers that may be used to access these data. Or, if the data are not a core part of the research being presented in your study, we ask that you remove the phrase that refers to these data. 

R4: We sincerely apologize for including the phrase “data not shown” in the manuscript. As the data for the Acetylene Reduction Assay (ARA) conducted using Yeast Manitol media is not a core part of the research, it is removed from the manuscript. The results of Acetylene Reduction Assay conducted using nitrogen free media (modified Fraquil medium) is the important core part of the research and hence the complete data related to that experiment is presented in the manuscript.

Additional Editor Comments (if provided):

C5: The manuscript is suitable for publication but needs some minor adjustments. Please make the changes indicated by the 2 external reviewers. In addition, I ask that you try to have the text edited by a native English speaker, as there are many grammatical errors. In addition, a reviewer commented that "more logical explanations which would definitely enhance the quality of this manuscript". Thus, I ask that you go through the Discussion section, making sure the argument flows logically. Perhaps add some subheadings to indicate topics being discussed and divide some of the longer paragraphs into a few shorter ones.

R5: We appreciate additional Editor Comments on our manuscript. As advised by the Editor, we incorporated all the changes indicated by the two external reviewers and changes are described under each reviewer comments. As mentioned under C3: the revised text is fully edited from professional editing service (www.editage.com). We believe the quality of this manuscript is now enhanced substantially. Furthermore, as suggested by the Editor, we thoroughly re-visited the Discussion section and revised; bring more logical discussion, added new references particularly under the sub-section “Diversity in free-living nitrogen-fixing bacterial community”. As suggested, the whole Discussion section is now divided into four sub-sections which reflected better about the topics being discussed and also divided some of the longer paragraphs into shorter ones as suggested. 

Review Comments to the Author:

Reviewer #1: 

C1: At line no. 27 you have used BNF as abbreviation for Biological nitrogen fixation. However, at line no 74 you are using same abbreviation for Biological nitrogen fixers. Please correct this, both biological nitrogen fixation and biological nitrogen fixers are absolutely different.

R1: Indeed, BNF is the abbreviation for Biological Nitrogen Fixation. Therefore, the BNF in line no. 74 was corrected to ‘biological nitrogen fixers’. We deeply apologize for the confusion. 

C2: At line no 116, please explain the methodology of rhizospheric soil sample collection in details.

R2: As per reviewer’s advice, the methodology for soil and root nodule sample collection was revised completely in detail and incorporated under the section “Sampling” in the manuscript. 

C3: Please put space between number and °C throughout the manuscript. For example write 10 °C instead of 10°C.

R3: As per the reviewer’s advice, a space was added between the number and °C throughout the manuscript.

C4: Please make the tables compact. It would be easy to understand data if more information is present in single page rather than the multiple pages.

R4: We agree with the reviewer that it is easy to understand the data if more information is present in a single page rather than the multiple pages. We tried our best to compress the table as suggested. Consequently, the two tables were compressed in to four pages from the original eight pages and are presented in the manuscript.

C5: In line no 434 please replace N2 with N2.

R5: We apologize for the mistake in line no. 434. The N2 in line no. 434 was replaced with N2. Also ‘N2’ was checked throughout the manuscript and corrected.

C6: At line no 461 the authors have written “In this investigation, a structure resembles to root nodule often found in Leguminaceae plant roots was observed and assumed similar to typical nitrogen-fixing root nodules produced by Rhizobium sp.”. How could you make this statement without showing results of nodule characterization? Please justify the about statement written in the manuscript.

R6: We intensely appreciate the reviewer’s concern on nodule characterization. In order to justify the statement “In this investigation, a structure resembles to root nodule often found in Leguminaceae plant roots was observed and assumed similar to typical nitrogen-fixing root nodules produced by Rhizobium sp.”, the picture of root nodule documented during the root nodule sample collection is added in the supplementary material ( S2 Fig) for reference. 

C7: Image quality of figures is very poor. Please incorporate high quality images.

R7: As per the reviewer’s advice, high quality of the figures with 300dpi resolution were produced and incorporated in the manuscript during the resubmission. We hope the current figures are better quality. 

C8: This study is showing that Agrobacterium tumefaciens, which has been renamed as Rhizobium radiobacter, was recovered by root nodules and have potential to fix atmospheric nitrogen outside of the host as shown in ARA based analysis. However, other nodules forming Rhizobium species does not fix nitrogen outside of the host. Moreover, characterization of the bacterial isolates is only based on the partial 16S rDNA sequencing which do not have sufficient resolution to characterize closely related bacterial species. Therefore, please explain your results more clearly and logically. Please go through the following article as it can help you to explain your results in more details:

Velázquez, E., Peix, A., Zurdo-Piñiro, J. L., Palomo, J. L., Mateos, P. F., Rivas, R., ... & Martínez-Molina, E. (2005). The coexistence of symbiosis and pathogenicity-determining genes in Rhizobium rhizogenes strains enables them to induce nodules and tumors or hairy roots in plants. Molecular plant-microbe interactions, 18(12), 1325-1332.

R8: We appreciate reviewer comments on the above concern. We re-visited the discussion section and tried to discuss our results more elaborately and therefore a completely new discussion is added with few new references, particularly under sub-sections: Diversity in free-living nitrogen-fixing bacterial community and Root nodules like structure of V. pachyceras. In order to discuss more clearly, the discussion section is now divided into four sub-sections, such as 

1. Diversity in free-living nitrogen-fixing bacterial community 

2. Assessment of nitrogen-fixing ability by acetylene reduction assay (ARA)

3. Root nodules like structure of V. pachyceras and A. tumefaciens 

4. A. tumefaciens as diazotrophs

Reviewer #2: 

C1: The authors enriched and isolated several number of free and symbiotic N2-fixers from rhizosphere and root nodes of four plant species in Kuwait semi-arid desert ecosystem. Apart from precipitation, nitrogen is important for plant growth and microbial nitrogen-fixers play essential roles in nitrogen supply for the semi-arid desert ecosystem. The isolation of many nitrogen-fixing bacteria can pave the way to fully understand N cycling in the special ecosystem, and it shows the most significance especially in the modern popular "metagenomics" era. The manuscript has been well organized and the materials and methods were well described. There was only minor parts needed to be revised: Discuss the differences in microbial compositions between this study and other studies/ ecosystems.

R1: We really appreciate reviewer overall comments on our manuscript. We thank reviewer for directing about bringing some discussion on the differences in microbial compositions between this study and other studies elsewhere. As suggested, we added some new discussions in the last two paragraphs under the sub-section “Diversity in free-living nitrogen-fixing bacterial community”. 

C2: L34-36, repetead sentences "In this study,......species"

R2: We sincerely apologize for the repetition of the sentences in L34-36. The repeated sentences were removed and revised in the manuscript.

C3: L84, precipitation is also important

R3: We agree with the comment. As suggested, we revised the sentence and added about the precipitation information with a new reference. 

C4: L375, references?

R4: As suggested by the reviewer, we added new references to support the statement. 

C5: L376, Are there any differences in microbial compositions between this study and other studies/ ecosystems? What's the difference?

R5: Yes, there are differences in microbial compositions between the ecosystems. To address reviewer concern, we added some new discussions in the last two paragraphs under the sub-section “Diversity in free-living nitrogen-fixing bacterial community”. 

We think that the manuscript has been greatly improved by these revisions including English language editing by the professional and we hope that you will now find it suitable for publication in the PLOS ONE Journal.

---

## [Editor Report · Decision Letter 2]

20 Nov 2019

Divulging diazotrophic bacterial community structure in Kuwait desert ecosystems and their N2-fixation potential

PONE-D-19-20527R2

Dear Dr. Quoreshi,

We are pleased to inform you that your manuscript has been judged scientifically suitable for publication and will be formally accepted for publication once it complies with all outstanding technical requirements.

With kind regards,

Andrew R. Zimmerman, PhD

Academic Editor

PLOS ONE
---

## [Editor Report · Acceptance letter]

2 Dec 2019

PONE-D-19-20527R2 

Divulging diazotrophic bacterial community structure in Kuwait desert ecosystems and their N2-fixation potential 

Dear Dr. Quoreshi:

I am pleased to inform you that your manuscript has been deemed suitable for publication in PLOS ONE. Congratulations! Your manuscript is now with our production department. 

With kind regards,

on behalf of

Dr. Andrew R. Zimmerman 

Academic Editor

PLOS ONE